# Comprehensive Multimorbidity Patterns in Older Patients Are Associated with Quality Indicators of Medication—MoPIM Cohort Study

**DOI:** 10.3390/ijerph192315902

**Published:** 2022-11-29

**Authors:** Marina Lleal, Marisa Baré, Sara Ortonobes, Daniel Sevilla-Sánchez, Rosa Jordana, Susana Herranz, Maria Queralt Gorgas, Mariona Espaulella-Ferrer, Marta Arellano, Marta de Antonio, Gloria Julia Nazco, Rubén Hernández-Luis

**Affiliations:** 1Institutional Committee for the Improvement of Clinical Practice Adequacy, Clinical Epidemiology and Cancer Screening Department, Parc Taulí Hospital Universitari, Institut d’Investigació i Innovació Parc Taulí (I3PT), 08208 Sabadell, Catalonia, Spain; 2Department of Paediatrics, Obstetrics and Gynaecology, Preventive Medicine and Public Health, Autonomous University of Barcelona (UAB), 08193 Bellaterra, Catalonia, Spain; 3Research Network on Health Services in Chronic Patients (REDISSEC), ISCIII, 28029 Madrid, Spain; 4Research Network on Chronicity, Primary Care and Health Promotion (RICAPPS), ISCIII, 28029 Madrid, Spain; 5Pharmacy Department, Parc Taulí Hospital Universitari, Institut d’Investigació i Innovació Parc Taulí (I3PT), 08208 Sabadell, Catalonia, Spain; 6Pharmacy Department, Parc Sanitari Pere Virgili, 08023 Barcelona, Catalonia, Spain; 7Internal Medicine Department, Parc Taulí Hospital Universitari, Institut d’Investigació i Innovació Parc Taulí (I3PT), 08208 Sabadell, Catalonia, Spain; 8Acute Care Geriatric Unit, Parc Taulí Hospital Universitari, Institut d’Investigació i Innovació Parc Taulí (I3PT), 08208 Sabadell, Catalonia, Spain; 9Geriatrics Department, Consorci Hospitalari de Vic, 08500 Vic, Catalonia, Spain; 10Geriatrics Department, Consorci Parc de Salut MAR, 08003 Barcelona, Catalonia, Spain; 11Pharmacy Department, Consorci Parc de Salut MAR, 08003 Barcelona, Catalonia, Spain; 12Pharmacy Department, Hospital Universitario de Canarias, 38320 La Laguna, Canarias, Spain; 13Internal Medicine Department, Hospital Universitario de Canarias, 38320 La Laguna, Canarias, Spain

**Keywords:** older patient, multimorbidity, cluster analysis, polypharmacy, potentially inappropriate medication, potential prescribing omission, adverse drug reaction, healthcare quality indicator

## Abstract

Multimorbidity is increasing and poses a challenge to the clinical management of patients with multiple conditions and drug prescriptions. The objectives of this work are to evaluate if multimorbidity patterns are associated with quality indicators of medication: potentially inappropriate prescribing (PIP) or adverse drug reactions (ADRs). A multicentre prospective cohort study was conducted including 740 older (≥65 years) patients hospitalised due to chronic pathology exacerbation. Sociodemographic, clinical and medication related variables (polypharmacy, PIP according to STOPP/START criteria, ADRs) were collected. Bivariate analyses were performed comparing previously identified multimorbidity clusters (osteoarticular, psychogeriatric, minor chronic disease, cardiorespiratory) to presence, number or specific types of PIP or ADRs. Significant associations were found in all clusters. The osteoarticular cluster presented the highest prevalence of PIP (94.9%) and ADRs (48.2%), mostly related to anxiolytics and antihypertensives, followed by the minor chronic disease cluster, associated with ADRs caused by antihypertensives and insulin. The psychogeriatric cluster presented PIP and ADRs of neuroleptics and the cardiorespiratory cluster indicators were better overall. In conclusion, the associations that were found reinforce the existence of multimorbidity patterns and support specific medication review actions according to each patient profile. Thus, determining the relationship between multimorbidity profiles and quality indicators of medication could help optimise healthcare processes. Trial registration number: NCT02830425.

## 1. Introduction

The clinical management of older patients with multiple conditions and pharmaceutical treatments poses a big challenge for healthcare professionals and systems. On top of the ageing population, a high prevalence of multimorbidity (classically defined as the presence of two or more coexisting chronic conditions) has been described worldwide and is expected to continue increasing [1,2]. Therefore, conducting research on how to improve the care of multimorbid patients in healthcare services that have traditionally focused on single diseases should be deemed of the utmost priority [3,4].

Various definitions and research methodologies have been developed in order to shed light on the concept of multimorbidity, and there is accumulating evidence suggesting that chronic conditions give rise to association patterns [5,6,7]. Although there is no standard yet, many publications have successfully identified multimorbidity patterns [8,9], some of which are repeatedly found among different studies [10]. Furthermore, some patterns have been associated with outcomes such as lower function, higher healthcare utilisation, poor prognosis or higher mortality [11,12,13,14,15,16]. Therefore, identifying multimorbidity patterns could help design new strategies and guidelines focusing on the most appropriate practices according to each patient profile.

This is remarkably important in older patients who, in addition to multimorbidity, present polypharmacy and age-related factors that can influence and hinder pharmacological prescribing. Some examples include physiological changes in pharmacokinetics and pharmacodynamics, cognitive impairment, functional difficulties or geriatric syndromes [17,18,19,20]. All in all, prescribing while balancing the benefits and risks becomes an arduous task.

In this scenario, a lot of attention has been brought to potentially inappropriate prescribing (PIP), which may occur in situations such as prescribing medications with potential benefits that do not outweigh the harms (particularly given the existence of safer alternatives); prescribing an inappropriate dose or duration or a duplicate drug; or omitting potentially beneficial medications. Several tools have been developed to identify PIP, such as the STOPP/START (Screening Tool of Older Person’s potentially inappropriate Prescriptions/Screening Tool to Alert doctors to Right Treatment) criteria, the most used and validated in European older adults [21]. These criteria include both potentially inappropriate medications (PIMs) and potential prescribing omissions (PPOs). While PPOs can prevent patients from taking essential medication, leading to risks and negative outcomes [22,23], PIMs are a well-known risk factor for adverse drug reactions (ADRs) [24,25,26]. ADRs are highly frequent in older patients and have been estimated to be responsible for 10–30% of hospital admissions [27,28] as well as to occur in 16% of already hospitalised older patients [29].

Taking into account all these considerations, it is plausible that certain multimorbidity patterns may present an association with specific PIP and/or ADRs. Importantly, there are no publications to date studying this inter-relationship in detail. Identifying associations could pave the way to optimising and focusing medication review actions and improve healthcare in these complex patients, reducing undesired outcomes. The present analyses are part of the MoPIM multicentre cohort study [30], which has various objectives regarding multimorbidity, PIP and ADRs in older patients hospitalised due to chronic condition exacerbation. A set of four multimorbidity patterns were identified in a previous publication [31]; thus, the objectives of this work are to evaluate if any of these previously identified multimorbidity patterns are associated with the presence, number or specific types of PIP or ADRs.

## 2. Methodology

### 2.1. Design and Setting

A multicentre prospective cohort study including older patients hospitalised at the internal medicine or geriatric services at five general teaching hospitals in three different regions of Spain between September 2016 and December 2018 was conducted. The detailed protocol was previously published [30]. For the purposes of the study, older patients (≥65 years old) admitted as a result of the exacerbation of their chronic pathology were included. Patients referred to home hospitalisation, admitted because of an acute process unrelated to any chronic disease or with a fatal outcome expected at the time of admission, were not included. No written informed consent was deemed necessary for this study, according to the independent ethics committee.

### 2.2. Data Acquisition and Variables

The following sociodemographic and clinical data were retrieved from the electronic health records by the clinical team responsible for the patient: patient’s code, date of birth, sex, functional status just before admission (Barthel Index) [32], household (alone, with relatives or other people, in a nursing home) and existence of any contact with healthcare services in the 3 months prior to hospitalisation due to exacerbation of any chronic disease and destination at discharge from the present episode of hospitalisation (home, transfer to another hospital, transfer to a nursing home, death). Chronic active conditions were recorded from a consensual list of 64 conditions containing all chronic diseases of the Charlson Comorbidity Index [33] and including some risk factors as well. Geriatric syndromes and risk factors were also recorded from a list of 15 (see protocol [30]).

The number of chronic medications in the electronic prescription at the time of admission and the STOPP/START criteria (version 2) [21] detected upon admission, with the active principle involved, were collected by the pharmacist of the team. The 2nd version of STOPP/START criteria consists of a list of 114 medication indications, developed using a Delphi method by experts from different disciplines, who carried out a literature review. The criteria are directed to prevalent diseases in older patients, are ordered by physiological systems and are easy to relate to active diagnoses. This medication review process was part of the usual patient care routine in all participating centres. Medication was only considered chronic if prescribed at least 3 months before admission, and creams, ointments, healing materials and over-the-counter medicines were not considered. Active principles were considered individually when registering STOPP/START criteria, regardless of the administered drug combinations.

Finally, ADRs were identified by the clinical team both at admission and during the course of stay. ADRs were considered according to the WHO and the European Medicines Agency criteria [34,35,36]. The active principle involved and whether the ADR occurred at admission or during hospitalisation were collected. Consequences in terms of health (death, life-threatening, lengthening of hospitalisation, other important consequences under medical criteria) were registered if the ADR appeared during hospital stay.

### 2.3. Sampling and Analysis

Patients included were proportionally distributed to the annual volume of hospitalisations at the internal medicine and/or geriatric services of each centre.

The Updated Charlson Comorbidity Index [37] was calculated, adjusted by age and categorised by tertiles (2–6, 7–8, 9–14).

Multimorbidity patterns were identified using a soft clustering algorithm, as thoroughly described [31]. Firstly, some chronic conditions were grouped according to clinical criteria and then filtered by <2% prevalence, resulting in a list of 40 chronic conditions and 15 geriatric syndromes. Chronic conditions were weighted according to the required clinical management. Then, transformation and dimensionality reduction for the dataset were carried out with the PCAmix algorithm [38], and cluster analysis was performed with the fuzzy c-means algorithm [39]. This technique allowed for obtaining clusters of patients based on their chronic conditions and geriatric syndromes with a membership probability to every cluster. After computing several validation indexes [31], a range of statistically significant possibilities were obtained, and, after clinical revision and discussion among the research team, an eventual set of four clusters was established. These clusters were named ‘osteoarticular’, ‘psychogeriatric’, ‘minor chronic disease’ and ‘cardiorespiratory’. Patients were assigned to the cluster where their membership probability was highest.

All STOPP/START criteria were assessed, except for START criteria I (vaccines) due to difficulties of some centres in accessing the information. Regarding the implicit criterion STOPP A1 and given its high frequency, it was divided into the following categories according to the active principle involved: proton pump inhibitors, hypolipidemics, analgesics, acetylsalicylic acid, antihypertensives and others [40].

Active principles involved in ADRs were categorised in the following drug families: analgesic, angiotensin-converting enzyme (ACE) inhibitor, angiotensin II receptor blocker, antiarrhythmic, antibiotic, anticoagulant, antidepressant, antiepileptic, antiplatelet, antipsychotic, antivitamin K, benzodiazepine, beta blocker, bronchodilator, corticoid, loop diuretic, neuroleptic, insulin, opioid, oral anticoagulant, oral antidiabetic, potassium sparing diuretic, proton pump inhibitor, statin, thiazide diuretic and others. Equivalence with ATC (Anatomical Therapeutic Chemical) codes can be found in Appendix A.

Binary variables were created to describe the presence of any STOPP/START PIP, any STOPP PIM, any START PPO, any ADR, any ADR at admission and any ADR during hospitalisation. This was performed similarly with numerical variables for the number of STOPP/START PIP (excluding implicit criteria STOPP A1, A2, A3), number of STOPP PIM (excluding STOPP A1, A2, A3), number of START PPO, number of ADR, number of ADR at admission and number of ADR during hospitalisation.

Descriptive analyses were performed for all variables. Bivariate analyses were conducted to assess possible associations between multimorbidity clusters and the presence or type of PIP or ADR by the Fisher’s exact test. Most frequent PIP criteria were selected with the aim of analysing at least the top 10 criteria for PIMs and PPOs, resulting in a cut-off of 5% of patients of a cluster for STOPP criteria and 3% of patients of a cluster for START criteria. ADRs were only analysed if present in at least 5 patients of a cluster. Post hoc pairwise Fisher’s exact tests were conducted for those previously significant tests (*p* < 0.05), and *p*-values were corrected for multiple hypothesis testing using Benjamini–Hochberg false discovery rate (FDR) method [41] at a 5% cut-off.

Comparisons between number of PIP or ADR were performed using the Kruskal–Wallis test. Pairwise comparisons between distributions of the number of PIP or ADR among the different clusters were performed using the Kolmogorov–Smirnov test. These comparisons were performed for the following variables: number of STOPP/START PIP (excluding implicit criteria, i.e., STOPP A), number of STOPP PIMs (excluding implicit criteria, i.e., STOPP A), number of START PPOs, number of ADRs, number of ADRs at the time of admission and number of ADRs during hospitalisation.

All analyses were performed with R (R Foundation for Statistical Computing, Vienna, Austria) [42].

## 3. Results

### 3.1. Sociodemographic and Clinical Characteristics of the Cohort

A total of 740 patients were included; 53.2% were females, and 98.7% were diagnosed with two or more chronic conditions. The mean age was 84.1 (SD 7.0) years, and the mean Barthel Index was 65 (median 75). The cardiorespiratory cluster contained most patients (*n* = 325, 43.9%), followed by the psychogeriatric (*n* = 151, 20.4%), osteoarticular (*n* = 137, 18.5%) and minor chronic disease (*n* = 127, 17.2%) clusters. Sociodemographic and clinical variables are summarised in Table 1, according to the assigned multimorbidity cluster. The prevalences of chronic conditions and geriatric syndromes are described according to multimorbidity cluster in Appendix A. Among all the detected STOPP/START criteria, the most prevalent STOPP criteria were A1: Drugs prescribed without an evidence-based clinical indication (*n* = 310, 25.7%), D5: Benzodiazepines for ≥4 weeks (*n* = 247, 20.5%) and K1: Benzodiazepines (*n* = 131, 10.9%), from a total of 1206 criteria detected. The most prevalent START criteria were E5: Vitamin D supplement in older people who are housebound or experiencing falls or with osteopenia (*n* = 76, 21.5%), H2: Laxatives in patients receiving opioids regularly (*n* = 50, 14.1%) and A8: Appropriate beta-blocker with stable systolic heart failure (*n* = 39, 11.0%), from a total of 353 criteria detected.

### 3.2. Relationship between Multimorbidity Clusters and Potentially Inappropriate Prescribing

A bivariate analysis was performed to test the association of belonging to a cluster and presenting any STOPP/START PIP, STOPP PIMs or START PPOs. Table 2 shows that a significant association was found in all these three variables. Pairwise comparisons (Appendix A) showed that the osteoarticular cluster was significantly different from all others regarding the presence of any PIP or any PIMs, and together with the minor chronic disease were significantly different in presence of PPOs from the other two.

Next, we compared the number of PIP, PIMs and PPOs between clusters taking into account only explicit criteria. Differences were found between clusters in all three variables (Kruskal–Wallis test: *p* < 0.001 in number of PIP, *p* < 0.001 in number of PIMs, *p* = 0.001 in number of PPOs). Pairwise comparisons between distributions showed that the osteoarticular cluster in both the number of STOPP/START PIP and STOPP PIMs was different from the other clusters (Kolmogorov–Smirnov test: *p* < 0.001 and *p* < 0.005, respectively), meaning that patients of this cluster tend to present a larger number of PIP and PIMs (Appendix A). No significant differences were found in the distribution of the number of START PPOs.

Then, a bivariate analysis was performed focusing on certain specific criteria, selecting the most frequent ones. Significant associations were found in the STOPP PIM criteria related to benzodiazepines (STOPP D5, G5, K1) or ACE inhibitors/angiotensin receptor blockers (STOPP B11), which were significantly higher in the osteoarticular cluster. PIMs related to proton pump inhibitors (extracted from STOPP A1) were significantly lower in the cardiorespiratory cluster than the rest. PIMs involving neuroleptic drugs (STOPP K2) were most prevalent in patients of the psychogeriatric cluster and least prevalent in those of the cardiorespiratory cluster (Figure 1A, Appendix A).

Regarding START PPO analysis, the most frequent criteria were selected and compared between clusters as well. PPOs involving beta blockers (START A8) were positively associated with the cardiorespiratory cluster, while the lack of vitamin D prescribing (START E5) was negatively associated with it. Prescribing the omission of laxatives (START H2) was significantly higher in the osteoarticular and minor chronic disease clusters with respect to the others (Figure 1B, Appendix A).

### 3.3. Relationship between Multimorbidity Clusters and Adverse Drug Reactions

A total of 376 ADRs were reported in 245 patients (33.1%), and 59.6% of those were detected at the time of admission in 153 patients (20.7%). Having an ADR was significantly associated with belonging to particular multimorbidity clusters. Almost half of the patients in the osteoarticular and minor chronic disease clusters suffered at least one ADR, and both clusters were significantly different from the psychogeriatric and cardiorespiratory. When separating ADRs into those detected at the time admission and those occurred during hospital stay, these two clusters also showed a significantly higher percentage of patients (Table 3 and Appendix A).

With respect to the number of ADRs between different clusters, significant differences were found when considering all ADRs, those detected at admission and those occurred during hospitalisation (Kruskal–Wallis test: *p* < 0.001 in all cases). Afterwards, a pairwise comparison of the distributions was performed too, which showed that the osteoarticular and the minor chronic disease clusters presented a different distribution from the psychogeriatric and cardiorespiratory clusters both regarding the total number of ADRs and those detected at the time of admission (Appendix A). No significant differences were found in the distribution of the number of ADRs occurred during hospital stay.

To determine the association of multimorbidity clusters to specific types of ADRs, a bivariate analysis was performed, similarly to the one involving the STOPP/START criteria. Figure 2 shows the percentage of patients that suffered an ADR of a certain drug family according to their assigned multimorbidity cluster, only considering those ADRs detected in at least five patients in a cluster. Patients belonging to the osteoarticular cluster suffered ADRs involving ACE inhibitors more frequently, as well as patients in the minor chronic disease cluster, which more frequently experienced ADRs related to angiotensin receptor blockers or insulin, with respect to the psychogeriatric or cardiorespiratory clusters. Furthermore, ADRs to neuroleptic drugs were more frequently suffered in psychogeriatric patients, and those involving diuretics were also associated with multimorbidity cluster belonging; however, no pairwise differences could be found (Appendix A).

Finally, the relationship between multimorbidity cluster and the consequences of ADRs during admission was also tested, and a significant association was found (Figure 3; Fisher’s Exact Test: *p*-value = 0.02). For example, 18.8% of patients in the osteoarticular cluster who suffered an ADR during admission faced a life-threatening situation, whereas this did not happen to any patients in the psychogeriatric cluster. Nevertheless, in the psychogeriatric cluster, most ADRs caused a lengthening of hospital stay. None of the ADRs were fatal.

## 4. Discussion

### 4.1. Main Important Results and Novelty

Our study successfully detected significant associations between multimorbidity patterns and specific PIMs, PPOs and ADRs, showing that patients in each multimorbidity cluster tend to present comparable health problems and, therefore, identifying patients with similar needs. The osteoarticular cluster displayed the worst situation regarding PIP and ADRs, particularly related to anxiolytics and antihypertensives. The psychogeriatric cluster, despite having the lowest number of chronic prescriptions, presented PIP of proton pump inhibitors and neuroleptics, with the latter also causing ADRs. The minor chronic disease cluster was associated with ADRs caused by antihypertensives and insulin, and the cardiorespiratory cluster showed fewer PIP and ADRs overall. Altogether, our results support the prioritisation of medication review in patients from the osteoarticular cluster, which accounted for the largest proportion of patients with PIP or ADRs, along with the most severe consequences of ADRs, followed by the minor chronic disease cluster.

This is, to our knowledge, the first study to consider and analyse the relationship of multimorbidity patterns with quality indicators of medication, such as PIP and ADRs. Our approach includes a novel methodology of defining multimorbidity in conjunction with an extensive set of explicit and implicit criteria (STOPP/START) regarding PIMs and PPOs and together with an exhaustive registration of ADRs, providing a unique dataset integrating this information. Significant associations were found between clusters when considering presence, number or specific PIMs, PPOs and ADRs, suggesting that these situations should be differently managed according to each particular patient profile.

### 4.2. Clinical Implications

The osteoarticular cluster not only presented the highest percentage of patients with at least a PIP or PIM but also a larger number of them than the other clusters. This could be partially explained due to a higher number of chronic prescriptions and is also consistent with the high prevalence found in the three benzodiazepine-related criteria (STOPP D5, G5, K1), frequently coexisting in the same patients. We certainly expected an overall high prescription of benzodiazepines [43,44], especially in this cluster that has the highest prevalence of depression and anxiety (61.3%). This could even be a result of excessive medicalisation in a female-predominant cluster. It is well-known that benzodiazepine prescribing is excessive [45], and this situation becomes particularly concerning in this patient profile due to its association with negative outcomes such as falls, fractures, dependence and cognitive decline [46]. The almost-ubiquitous prevalence of chronic pain (92.7%), frailty (83.9%) and degenerative arthropathy (81.0%) stress the need for benzodiazepine deprescribing. Nonetheless, we acknowledge the complexity of this process.

Furthermore, we found that side effects in prescribing for pain management might not be properly addressed in the osteoarticular cluster. A significant association with STOPP L2 and START H2 criteria was found, both referring to the requirement of laxative prescription in patients under opioid therapy. This was also found in the minor chronic disease cluster, consistent with its high prevalence of patients with chronic pain (69.3%). Nevertheless, it is plausible that patients in both clusters, which are already taking a large number of medications, could be using some herbal or over-the-counter products. Besides, these results did not correlate well with the prevalence of constipation in the minor chronic disease cluster, which accounted for the lowest proportion (32.3%). This could be explained by differences in the involved opioids, with patients prescribed transdermal fentanyl having less risk of constipation than those on oxycodone or morphine [47,48].

Additionally, the use of antihypertensive and diuretic drugs also posed a challenge in patients from both the osteoarticular and the minor chronic disease clusters (with many registered ADRs) compared to the others. Although it was not always possible to establish significant pairwise comparisons due to a low number of cases, there was a significant association overall. In the specific case of ACE inhibitors, they were found to be significantly different in the osteoarticular cluster as well as detected as PIMs in STOPP criteria B11. Conversely, angiotensin receptor blockers caused ADRs in a higher proportion of patients in the minor chronic disease cluster but were not labelled as inappropriately prescribed. Our results, therefore, suggest that, although side effects of antihypertensive drugs are well-known [49,50,51], decompensations particularly occur in these patients and may lead to life-threatening situations, which were consistently found to be higher in both clusters. However, these situations may be harder to address, as they might not always be identified as PIMs.

Remarkably, some of the ADRs detected were not caused by a previously identified PIM, revealing one of the limitations of PIP/PPO detection tools. This was especially evident in the minor chronic disease cluster, which unexpectedly presented a high number of ADRs. This cluster appears to be the most heterogeneous of all, and it is possible that this situation may have occurred due to a single disease prescribing approach, where various medications are accumulatively prescribed by different professionals. Moreover, it is also plausible that medication review in these patients, who are the least dependent and are mostly living with other people or alone, was not prioritised.

This situation contrasts with the findings regarding the psychogeriatric cluster, with the lowest number of chronic prescriptions, low number of PIP and ADRs and no in-hospital life-threatening ADRs. This could be explained by an increased effort in medication review and comprehensive clinical management. However, two situations stood out: PIMs of proton pump inhibitors were especially high, and neuroleptics were detected as PIMs and also caused ADRs. These results could be expected, yet problematic, as both are related to a variety of adverse outcomes [52,53,54,55,56,57], which could cause a high burden in already very frail patients. Therefore, there would still be room for deprescribing.

Lastly, patients in the cardiorespiratory cluster, containing almost half of the cohort, were undoubtedly in the most preferable situation: no remarkable PIMs or ADRs. Only one PPO criterion stood out: lack of a beta blocker prescription, which could be explained by the opposite effect of beta blockers to beta agonists, usually administered in patients with chronic obstructive pulmonary disease (COPD). However, the current literature recommends beta blocker prescribing in patients with heart disease and COPD [58]. All in all, these patients were minimally dependent, with a low number of chronic conditions and geriatric syndromes. Thus, it is plausible that although they had a chronic pathology exacerbation that led to hospital admission, their health status was overall better and only restricted to cardiorespiratory problems. Therefore, our results suggest these patients might be easier to handle.

Taken together, our results show how multimorbidity profiles built according to chronic conditions and geriatric syndromes have a significant association with PIP and ADRs. On the one hand, these results support the existing evidence on the concept of multimorbidity forming association patterns. The associations found on each profile to some extent validate our methodological approach, which successfully allocates patients with similar situations and needs. On the other hand, these results suggest that appropriate actions according to each patient profile could be taken in hospitals but also in primary care settings, which could be a useful approach to optimise and offer better health care. It is essential to effectively direct efforts in these complex patients, and, in this case, patient categorisation strategies together with multidisciplinary teams could be helpful to address the situation. Thus, further studies are needed to incorporate multimorbidity approaches in all levels of healthcare.

### 4.3. Comparison to Other Studies

Although there are previous studies on PIP and ADRs in different types of cohorts of multimorbid patients studying the risk factors, outcomes and interventions, the vast majority define multimorbidity with the presence of two or more chronic conditions or consider those from a short list [59,60,61,62,63]. These definitions, although easily determined, become too simplistic in a setting of older hospitalised patients, where almost all are multimorbid (98.7% in our cohort). Therefore, new analytical strategies need to be explored that consider a more comprehensive definition of multimorbidity, as there are currently no publications directly comparable to ours.

The most similar study is the one carried out by Teh et al., in 2018 [14], comparing multimorbidity patterns to the presence of any PIM or any PPO, but without considering explicit criteria nor ADRs. Multimorbidity profiles are built with a similar methodology and agreed upon by consensus among a multidisciplinary team involving clinicians as well. However, the cohort is comprised of community patients over 80 years old, considering a list of 14 conditions and using the first version of STOPP/START criteria, which disallows comparisons. Nonetheless, the cluster with the highest proportion of patients with any PIMs or PPOs is called ‘depression and arthritis’ and presents the highest prevalence of osteoporosis and osteoarthritis, suggesting that this cluster could be similar to the osteoarticular cluster from our cohort, which also presents depression and anxiety. In this article, the authors conclude that profiles of conditions may carry stronger associations with cross-sectional outcomes than the sum of those conditions.

### 4.4. Strengths and Limitations

This study presents multiple strengths. As a multicentre study, it presents increased external validity. Its prospective design ensures high data quality by an accurate and thorough registering of variables that may commonly be underreported, such as PIP and ADRs. In this sense, it is important that a multidisciplinary team composed of pharmacists and physicians work together in the medication-review process.

Furthermore, the approach of multimorbidity cluster analysis is novel and methodologically robust, incorporating both chronic conditions and geriatric syndromes and allowing for work with patient profiles instead of single conditions. Moreover, this study has been carried out with a well-defined cohort of hospitalised patients admitted due to chronic condition exacerbation. This constitutes a particularly vulnerable and complex group of patients, who may largely benefit from a reduction in negative outcomes. Finally, the unique approach of the study, considering multimorbidity patterns with specific PIP and ADRs all together, allows for obtaining a reliable picture of a complex situation that needs to be explored and addressed.

Nonetheless, some limitations of this study need to be taken into account. Firstly, methodological approaches to tackle multimorbidity are highly variable, and there is no consensus on the best practices to determine multimorbidity patterns. The results of this study are conditioned to the predefined multimorbidity patterns, which could be questioned, although they were comprehensive, considering a large list of conditions, and consensually selected by a research team including clinicians. Secondly, there may be differences in the reporting of PIP or ADRs between centres or healthcare professionals, despite the large efforts made to homogenise criteria. Regarding ADRs, active principles alone could not be analysed due to low occurrence so had to be collapsed into higher-level categories. Finally, the direct relationship between PIP and ADRs was not addressed, as this was far from the objectives of these analyses.

## 5. Conclusions

In older patients admitted to hospital because of chronic conditions’ exacerbation, it is possible to define multimorbidity clusters that are associated with quality indicators of medication prescribing such as the presence, number or specific types of PIP and ADRs. These associations validate and support the existence of such clusters and point to specific prescriptions that could be primarily reviewed and made adequate for each patient profile. Thus, determining the relationship between multimorbidity profiles and the quality indicators of medication could be key in remodelling and optimising healthcare processes in order to tackle the increasing prevalence of older patients with multimorbidity.

## Figures and Tables

**Figure 1 ijerph-19-15902-f001:**
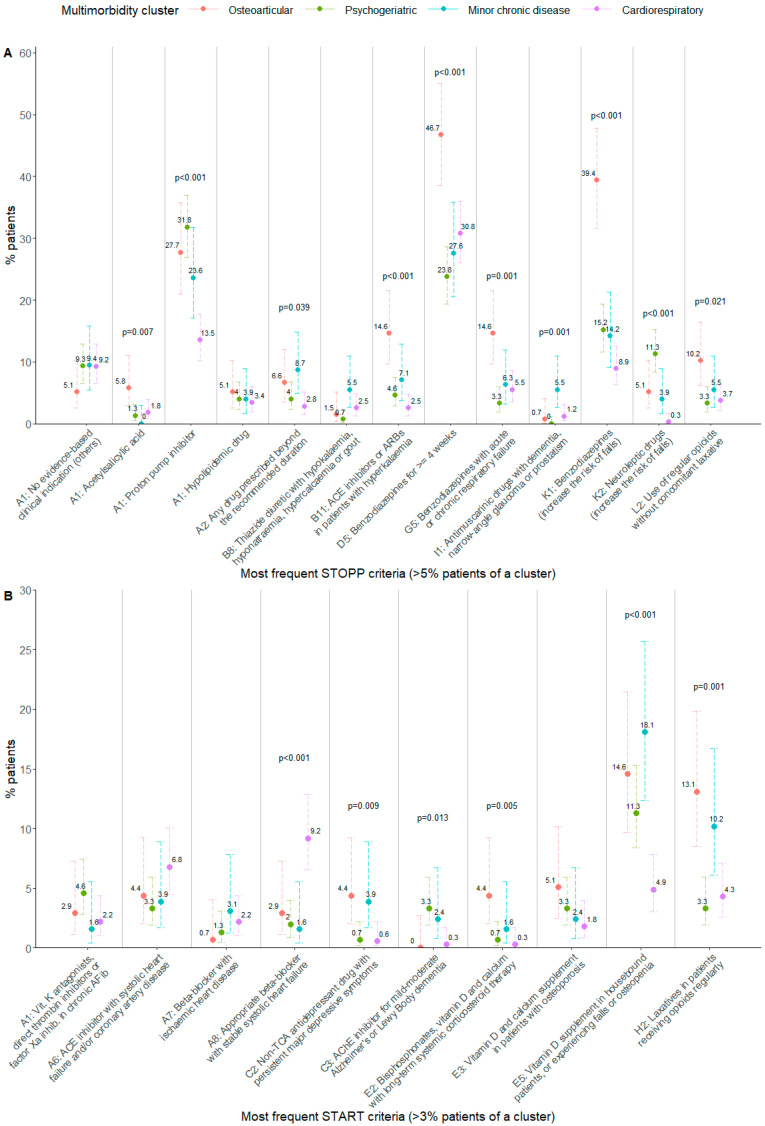
(**A**): Percentage of patients per cluster having the most frequent STOPP PIM criteria. (**B)**: Percentage of patients per cluster having the most frequent START PPO criteria. Fisher’s exact test *p*-value: *p*-values are shown in the figure when *p* < 0.05. Error bars show 95% confidence interval for the estimated proportion. ACE: angiotensin converting enzyme; ARB: angiotensin II receptor blocker; AFib: atrial fibrillation; TCA: tricyclic antidepressant; AChE: acetylcholinesterase.

**Figure 2 ijerph-19-15902-f002:**
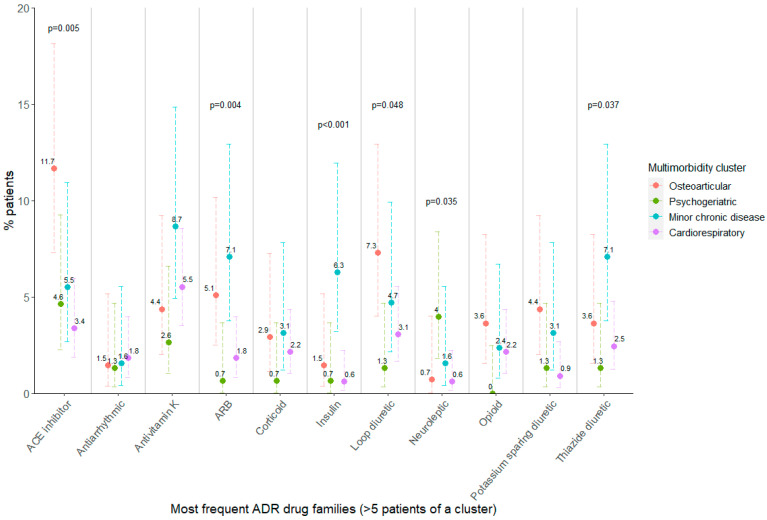
Percentage of patients per cluster having at least one ADR registered in the most frequent drug families. Fisher’s exact test *p*-value is shown when *p* < 0.05. ADR: adverse drug reaction; ACE: angiotensin-converting enzyme; ARB: angiotensin II receptor blocker. Error bars show 95% confidence interval for the estimated proportion.

**Figure 3 ijerph-19-15902-f003:**
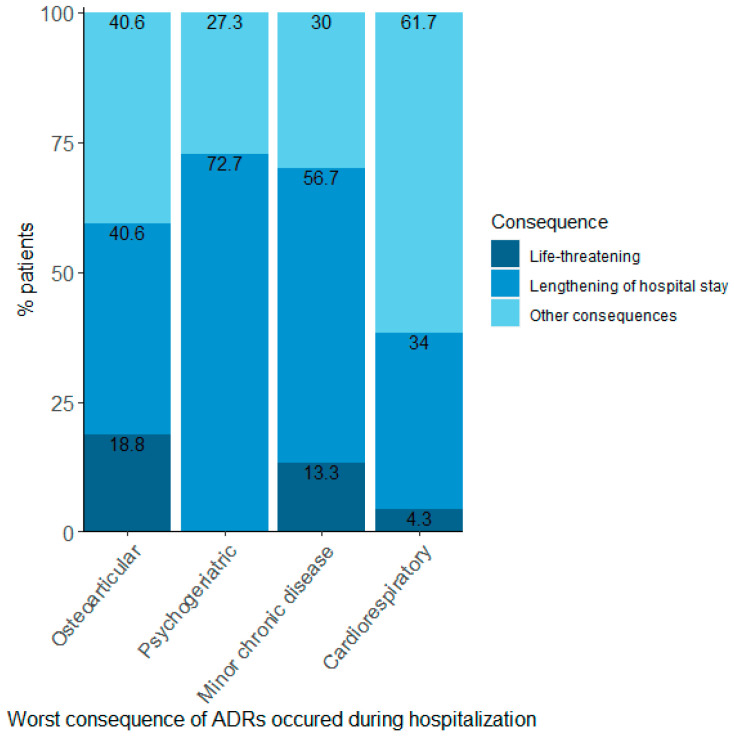
Proportions of patients per cluster according to the worst consequence suffered in those patients with an ADR during hospitalisation. ADR: adverse drug reaction. Fisher’s Exact Test: *p*-value = 0.02.

**Table 1 ijerph-19-15902-t001:** Sociodemographic and clinical variables of the cohort according to the assigned multimorbidity clusters.

		Osteo-Articular	Psycho-Geriatric	Minor Chronic Disease	Cardio-Respiratory
*n* (%)		137 (18.5)	151 (20.4)	127 (17.2)	325 (43.9)
Age at the time of admission (years, mean ± SD)		84.3 ± 6.3	85.1 ± 6.9	83.1 ± 7.2	84.1 ± 7.2
Sex, *n* (%)	Female	104 (75.9)	85 (56.3)	50 (39.4)	155 (47.7)
	Male	33 (24.1)	66 (43.7)	77 (60.6)	170 (52.3)
Barthel Index (mean ± SD)		61.6 ± 24.7	34.6 ± 31.4	77.4 ± 25.6	75.9 ± 27.2
No. of chronic pathologies (mean ± SD)		11.5 ± 3.6	7.7 ± 3.1	10.2 ± 3.1	7.2 ± 2.3
No. of geriatric syndromes (mean ± SD)		7.7 ± 1.8	9.1 ± 2.0	5.3 ± 2.8	4.2 ± 2.0
No. of chronic prescriptions (mean ± SD)		12.3 ± 4.58	9.5 ± 3.81	11.1 ± 4.0	10.1 ± 4.1
Updated Charlson Comorbidity Index, age-adjusted, *n* (%)	2–5	27 (19.7)	22 (14.6)	29 (22.8)	70 (21.5)
	6–8	77 (56.2)	87 (57.6)	62 (48.8)	185 (56.9)
	9–14	33 (24.1)	42 (27.8)	36 (28.3)	70 (21.5)
Household, *n* (%)	Alone	27 (19.7)	15 (9.9)	21 (16.5)	59 (18.2)
	Nursing home	17 (12.4)	35 (23.2)	8 (6.3)	35 (10.8)
	With relatives/other people	93 (67.9)	101 (66.9)	98 (77.2)	231 (71.1)
Chronic pathology exacerbation 3 months prior to admission, *n* (%)	No	26 (19.0)	37 (24.5)	30 (23.6)	132 (40.6)
	Yes	111 (81.0)	114 (75.5)	97 (76.4)	193 (59.4)
Destination at discharge, *n* (%)	Home	85 (62.0)	72 (47.7)	93 (73.2)	218 (67.1)
	Nursing home	18 (13.1)	35 (23.2)	13 (10.2)	39 (12.0)
	Another hospital	16 (11.7)	16 (10.6)	16 (12.6)	53 (16.3)
	Death	18 (13.1)	28 (18.5)	5 (3.9)	15 (4.6)

SD: standard deviation.

**Table 2 ijerph-19-15902-t002:** Presence of any PIP, PIMs or PPOs according to STOPP/START criteria in relation to the assigned multimorbidity clusters.

	Osteo-Articular	Psycho-Geriatric	Minor Chronic Disease	Cardio-Respiratory	*p*-Value
*n* (%)	137 (18.5)	151 (20.4)	127 (17.2)	325 (43.9)	
Any STOPP/START PIP	130 (94.9)	118 (78.1)	106 (83.5)	249 (76.6)	<0.001
Any STOPP PIMs	117 (85.4)	109 (72.2)	91 (71.7)	225 (69.2)	0.002
Any START PPOs	93 (67.9)	87 (57.6)	79 (62.2)	148 (45.5)	<0.001

Fisher’s exact test *p*-value is shown. PIP: potentially inappropriate prescribing, PIM: potentially inappropriate medication, PPO: potential prescribing omission.

**Table 3 ijerph-19-15902-t003:** Presence of any ADR in relation to the assigned multimorbidity clusters.

	Osteo-Articular	Psycho-Geriatric	Minor Chronic Disease	Cardio-Respiratory	*p*-Value
*n* (%)	137 (18.5)	151 (20.4)	127 (17.2)	325 (43.9)	
Any ADR	66 (48.2)	31 (20.5)	60 (47.2)	88 (27.1)	<0.001
Any ADR at admission	45 (32.8)	22 (14.6)	39 (30.7)	47 (14.5)	<0.001
Any ADR during hospitalisation	32 (23.4)	11 (7.3)	30 (23.6)	47 (14.5)	<0.001

Fisher’s exact test *p*-value is shown. ADR: adverse drug reaction.

## Data Availability

The data presented in this study are openly available in Zenodo at DOI 10.5281/zenodo.7371151.

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
