# Peer review of "Comprehensive Multimorbidity Patterns in Older Patients Are Associated with Quality Indicators of Medication—MoPIM Cohort Study"

_ijerph, 2022, doi:10.3390/ijerph192315902_

Round 1

Reviewer 1 Report

Dear Authors

I'm glad to have the opportunity to review the manuscript. It raises an important topic: multimorbidity in older patients.

In an aging society, this is an important and topical topic.

The presented data are interesting, but the manuscript needs a minor improvement.

In the methodology section, please add a description of the STOPP/START criteria to make it easier for the wider audience to understand the work.

Has the patient provided a medical and medication history, or is it available to healthcare professionals somehow? If the patient provided this information, there would be another study limitation. It is worth pointing out the context and describing this in the manuscript.

Author Response

Dear Authors

I'm glad to have the opportunity to review the manuscript. It raises an important topic: multimorbidity in older patients.

In an aging society, this is an important and topical topic.

We are thankful to the reviewer for their comments and appreciation of the importance of the topic addressed in the manuscript.

The presented data are interesting, but the manuscript needs a minor improvement.

In the methodology section, please add a description of the STOPP/START criteria to make it easier for the wider audience to understand the work.

We agree with the reviewer and have added a description of the STOPP/START criteria in the methods section 2.2., 3rd paragraph (lines 123-127) as well as the full name in the introduction (lines 78-79).

Has the patient provided a medical and medication history, or is it available to healthcare professionals somehow? If the patient provided this information, there would be another study limitation. It is worth pointing out the context and describing this in the manuscript.

We acknowledge this needed clarification. Clinical data was retrieved from the electronic health records of the patients by the clinical team. This has been added in the methods section 2.2., 1st paragraph (lines 110-111). Regarding chronic medication, this information was retrieved from the electronic prescription, as stated in line 120.

Reviewer 2 Report

Excellent methods with clear methods.

Some minor suggestions:

--can you edit the information in the horizontal axis in figure 1 (just to make it easier for readers)

--in the text can you write the top 3 STOPP and START criteria

--in the discussion can you add the importance of future studies in primary care?

Author Response

Excellent methods with clear methods.

Some minor suggestions:

--can you edit the information in the horizontal axis in figure 1 (just to make it easier for readers)

Although we are not completely sure of what the reviewer intended to point, we acknowledge that the text in the horizontal axis might be excessive and thus difficult to read. Therefore, we have shortened it with the aim of improving readability, hoping this is useful and this new version of Figure 1 addresses the reviewer’s concern.

--in the text can you write the top 3 STOPP and START criteria

This information has been added in the results section 3.1. (lines 200-207).

--in the discussion can you add the importance of future studies in primary care?

We agree with the reviewer that further studies should be done in the primary care setting, considering for example that the possible PIP may arise there, as well as deprescribing actions or social prescribing, which could differ among different multimorbidity patterns. In fact, we have an ongoing study with data from the community, regarding multimorbidity patterns and their relationship to various outcomes. Therefore, we are aware that incorporating this point of view and generating evidence with patients from the community is of utmost importance and kindly appreciate that the reviewer pointed it out; however, we consider that it falls far from the objectives of this article and its discussion. Nevertheless, we have made some changes in the last paragraph of the discussion section 4.2. (lines 389, 390 and 393, 394) to explicitly state that it is not only important to further study this topic in hospital settings but also in primary care. We hope this addresses the reviewer’s concern.

Reviewer 3 Report

I commend you on this work. Polypharmacy, PIP, and ADRs are of significant concern in older adults due to their high prevalence and morbidity/mortality. I appreciate your desire to evaluate these as they relate to specific multimorbidity patterns. This mirrors the patients many providers care for, given the aging of the population and increasing medical complexity. 

1. The introduction did an excellent job of presenting the background of this work. Ideally, I would have suggested including the Beers criteria as part of your evaluation but appreciate that the STOPP/START criteria are much more frequently used in Europe. 

2. The methodology is clearly presented. I appreciate that all the disease of the Charlson Comorbidity Index were included in addition to 15 geriatric syndromes. This much more clearly reflects the significant mortality of most older adults than many manuscripts in the literature looking at medication issues. The exclusion criteria are quite reasonable. My primary concern regarding this study is that over-the-counter medications were excluded. These are a major source of ADRs in older adults, including significantly anticholinergic medication such as over-the-counter sleep aids with diphenhydramine. While I do not think this should preclude publication of this work and it is unlikely that it can be added at this time, I would strongly encourage that all medications (supplements, herbal, over-the-counter) be included in any future work. 

3. The demographics of the patient population is similar to what is observed in many hospitals in developed countries. This makes with work very applicable for a wide audience. 

4. I commend the discussion. It does an excellent job of presenting the findings in an easily interpretable manner and the results can be used in consideration of one's own patient population. I agree with the authors that this work is novel due to including the relatively complex comorbidity clusters. 

I believe this work is important in both highlighting the significant issues with PIP and ADRs in older adults. However, this is well known and presented in the literature. What sits this apart is the correlation of these in different comorbidity clusters. I believe further work in this area would be beneficial including if there is an impact of pharmacists in decreasing ADRs in the hospital if these clusters are taken into account and by also considering the impact of all over-the-counter medications in this type of complex, older population. 

The punctuation in this manuscript requires careful review. Several commas are misplaced and there are some minor issues with the English. 

Author Response

I commend you on this work. Polypharmacy, PIP, and ADRs are of significant concern in older adults due to their high prevalence and morbidity/mortality. I appreciate your desire to evaluate these as they relate to specific multimorbidity patterns. This mirrors the patients many providers care for, given the aging of the population and increasing medical complexity. 

We are thankful to the reviewer for their insightful comments and positive assessment, as well as the appreciation of the importance of our line of work.

  1. The introduction did an excellent job of presenting the background of this work. Ideally, I would have suggested including the Beers criteria as part of your evaluation but appreciate that the STOPP/START criteria are much more frequently used in Europe. 

We thank the reviewer for their appreciation of our work. As argued in the manuscript, the STOPP/START criteria are the most used and validated in Europe. Furthermore, since this is an observational study, the medication review process performed as part of the usual patient care routine in the participating centres was based on STOPP/START criteria.

  1. The methodology is clearly presented. I appreciate that all the disease of the Charlson Comorbidity Index were included in addition to 15 geriatric syndromes. This much more clearly reflects the significant mortality of most older adults than many manuscripts in the literature looking at medication issues. The exclusion criteria are quite reasonable. My primary concern regarding this study is that over-the-counter medications were excluded. These are a major source of ADRs in older adults, including significantly anticholinergic medication such as over-the-counter sleep aids with diphenhydramine. While I do not think this should preclude publication of this work and it is unlikely that it can be added at this time, I would strongly encourage that all medications (supplements, herbal, over-the-counter) be included in any future work. 

We are thankful for the reviewer’s comments regarding the methodology of the manuscript. We are well aware that over-the counter medications are important and could be involved in ADRs; however, their identification and dosage quantification becomes an arduous task, as this information is not usually registered in the electronic health records. Despite including these medications was not in the objectives of this work, we kindly appreciate the suggestion and will consider so in future studies.

  1. The demographics of the patient population is similar to what is observed in many hospitals in developed countries. This makes with work very applicable for a wide audience. 
  2. I commend the discussion. It does an excellent job of presenting the findings in an easily interpretable manner and the results can be used in consideration of one's own patient population. I agree with the authors that this work is novel due to including the relatively complex comorbidity clusters. 

I believe this work is important in both highlighting the significant issues with PIP and ADRs in older adults. However, this is well known and presented in the literature. What sits this apart is the correlation of these in different comorbidity clusters. I believe further work in this area would be beneficial including if there is an impact of pharmacists in decreasing ADRs in the hospital if these clusters are taken into account and by also considering the impact of all over-the-counter medications in this type of complex, older population. 

We truly appreciate the reviewer's insights and absolutely agree that this line of work should continue to explore if new ways of organising healthcare according to multimorbidity rather than single pathologies may have a beneficial impact in these complex, older patients.

The punctuation in this manuscript requires careful review. Several commas are misplaced and there are some minor issues with the English. 

We have noted this comment and carefully reviewed punctuation and grammar.